# Energy Balance and Risk of Mortality in Spanish Older Adults

**DOI:** 10.3390/nu13051545

**Published:** 2021-05-04

**Authors:** Camille Lassale, Álvaro Hernáez, Estefanía Toledo, Olga Castañer, José V. Sorlí, Jordi Salas-Salvadó, Ramon Estruch, Emilio Ros, Ángel M. Alonso-Gómez, José Lapetra, Raquel Cueto, Miquel Fiol, Lluis Serra-Majem, Xavier Pinto, Alfredo Gea, Dolores Corella, Nancy Babio, Montserrat Fitó, Helmut Schröder

**Affiliations:** 1Hospital del Mar Medical Research Institute (IMIM), 08003 Barcelona, Spain; ocastaner@imim.es (O.C.); mfito@imim.es (M.F.); HSchoeder@imim.es (H.S.); 2CIBER of Pathophysiology of Obesity and Nutrition (CIBEROBN), Instituto de Salud Carlos III, 28029 Madrid, Spain; etoledo@unav.es (E.T.); Sorli@uv.es (J.V.S.); jordi.salas@urv.cat (J.S.-S.); restruch@clinic.cat (R.E.); eros@clinic.cat (E.R.); angelmago13@gmail.com (Á.M.A.-G.); joselapetra543@gmail.com (J.L.); rcueto@uma.es (R.C.); miguel.fiol@ssib.es (M.F.); lluis.serra@ulpgc.es (L.S.-M.); xpinto@bellvitgehospital.cat (X.P.); ageas@unav.es (A.G.); dolores.corella@uv.es (D.C.); nancy.babio@urv.cat (N.B.); 3August Pi i Sunyer Biomedical Research Institute (IDIBAPS), 08036 Barcelona, Spain; 4Blanquerna School of Life Sciences, Universitat Ramon Llull, 08025 Barcelona, Spain; 5Centre for Fertility and Health, Norwegian Institute of Public Health, 0473 Oslo, Norway; 6Department of Preventive Medicine and Public Health, University of Navarra, 31008 Pamplona, Spain; 7IdiSNA, Navarra Institute for Health Research, 31008 Pamplona, Spain; 8Department of Preventive Medicine, Universidad de Valencia, 46010 Valencia, Spain; 9Unitat de Nutrició Humana, Departament de Bioquimica i Biotecnologia, Universitat Rovira i Virgili, 43201 Reus, Spain; 10Institut d’Investigació Sanitaria Pere Virgili (IISPV), 43204 Reus, Spain; 11Internal Medicine Service, Hospital Clínic, 08036 Barcelona, Spain; 12Lipid Clinic, Endocrinology and Nutrition Service, Hospital Clínic, 08036 Barcelona, Spain; 13Bioaraba Health Research Institute, Osakidetza Basque Health Service, Araba University Hospital, University of the Basque Country UPV/EHU, 01009 Vitoria-Gasteiz, Spain; 14Department of Family Medicine, Research Unit, Distrito Sanitario Atención Primaria Sevilla, 41013 Sevilla, Spain; 15Department of Preventive Medicine and Public Health, Universidad de Málaga, 29071 Málaga, Spain; 16Health Research Institute of the Balearic Islands (IdISBa), Hospital Son Espases, 07120 Palma de Mallorca, Spain; 17Instituto de Investigaciones Biomédicas y Sanitarias, Universidad de Las Palmas de Gran Canaria, 35016 Las Palmas, Spain; 18Centro Hospitalario Universitario Insular Materno Infantil, Servicio Canario de Salud, 35016 Las Palmas, Spain; 19Hospital Universitario de Bellvitge, 08907 L’Hospitalet de Llobregat, Spain; 20CIBER Epidemiology and Public Health (CIBERESP), Instituto de Salud Carlos III, 28029 Madrid, Spain

**Keywords:** energy balance, mortality, epidemiology

## Abstract

Clinical data on the direct health effects of energy deficit or surplus beyond its impact on body weight are scarce. We aimed to assess the association with all-cause, cardiovascular and cancer mortality of (1) sustained energy deficit or surplus, calculated according to each individual’s en-ergy intake (EI) and theoretical energy expenditure (TEE), and (2) mid-term change in total EI in a prospective study. In 7119 participants in the PREDIMED Study (*PREvención con DIeta MEDi-terránea*) with a mean age of 67 years, energy intake was derived from a 137-item food frequency questionnaire. TEE was calculated as a function of age, sex, height, body weight and physical ac-tivity. The main exposure was the proportion of energy requirement covered by energy intake, cumulative throughout the follow-up. The secondary exposure was the change in energy intake from baseline. Cox proportional hazard models were used to estimate hazard ratios and 95% con-fidence intervals for all-cause, cardiovascular and cancer mortality. Over a median follow-up of 4.8 years, there were 239 deaths (excluding the first 2 years). An energy intake exceeding energy needs was associated with an increase in mortality risk (continuous HR_10% over energy needs_ = 1.10; 95% CI 1.02, 1.18), driven by cardiovascular death (HR = 1.26; 95% CI 1.11, 1.43). However, consum-ing energy below estimated needs was not associated with a lower risk. Increments over time in energy intake were associated with greater all-cause mortality (HR_10% increase_ = 1.09; 95% CI 1.02, 1.17). However, there was no evidence that a substantial negative change in energy intake would reduce mortality risk. To conclude, in an older Mediterranean cohort, energy surplus or increase over a 5-year period was associated with greater risk of mortality, particularly cardiovascular mortality. Energy deficit, or reduction in energy intake over time were not associated with mortal-ity risk.

## 1. Introduction

Energy balance is thought to be a key element of health. Obesity, likely a result of energy excess, has been consistently associated with a greater risk of mortality [1] and the development of cardiometabolic diseases [2] and certain cancers [3]. However, at the other end of the spectrum, the direct effects on mortality of long-term calorie restrictions are scarce in human studies. In non-human primates, specifically rhesus monkeys, studies report that calorie restriction of 30% is associated with a reduction in age-related all-cause mortality [4], although a large study did not find clear differences between the calorie restriction and control animals [5]. A time series study in Cuba showed that a 35% decrease in energy intake (EI) (together with an increase in the physically active population to twice the baseline number) led to a 50% decline in diabetes-related deaths and a 20–35% decrease in cardiovascular disease-related deaths [6]. Long-term calorie restrictions have been associated with decreases in the development of cardiovascular risk factors [7,8], and of diabetes [9] and some evidence suggests anti-cancer mechanisms in humans [10,11]. However, no clear preventive effect of a real-life energy deficit on mortality in healthy individuals has been reported in humans. Only one report from the Malmö Diet and Cancer Study found that low calorie consumers did not have lower mortality than average or high calorie consumers [12]. Moreover, the effect of whether a person is consistently consuming energy over or under their total requirements has not been linked with differences in the incidence of chronic diseases to date.

The main objective was to evaluate the association with mortality of sustained energy deficit or surplus, calculated according to each individual’s EI and theoretical energy expenditure (TEE). A secondary objective was to assess the association between a change in total EI over a few years and all-cause, cardiovascular and cancer mortality.

## 2. Methods

### 2.1. Study Population

The study population was participants in the PREDIMED Study (*PREvención con DIeta MEDiterránea*). It was a large-scale, multicenter, randomized, 3-armed, controlled, intervention trial conducted in Spain between 2003 and 2010, aiming to assess the long-term effects of following a Mediterranean diet on the primary prevention of cardiovascular outcomes in a population at high cardiovascular risk [13,14]. Eligible volunteers were 55–80-year-old men and 60–80-year-old women free of cardiovascular disease presenting type 2 diabetes or at least three out of six factors: tobacco use, hypertension, high levels of low-density lipoprotein cholesterol, low concentrations of high-density lipoprotein cholesterol, overweight/obesity, and family history of premature coronary heart disease. The protocol of the study complies with the Declaration of Helsinki, was endorsed by local institutional ethic committees at all study sites, was registered with the International Standard Randomized Controlled Trial Number ISRCTN35739639 (http://www.isrctn.com/ISRCTN35739639 last accessed 3 May 2021), is available on the PREDIMED study website (http://www.predimed.es last accessed 3 May 2021), and has been described in previous publications [13,14]. All participants provided written informed consent. The dietary intervention, with a Mediterranean diet enriched with extra virgin olive oil, or nuts, and the control group—who received advice to follow a low-fat diet—have been extensively described elsewhere [14].

For the present analysis, we used PREDIMED as a cohort, and all analyses were adjusted for the intervention group. Of the 7447 randomized participants, we excluded 87 with no available data on EI, physical activity or on adherence to a Mediterranean diet at baseline. Moreover, we excluded extreme values of EI < 500 kcal/day for women or <800 kcal/day for men, or >3500 kcal/day for women or >4000 kcal/day for men, at baseline (*N* = 152) and each follow-up yearly visit [15]. Finally, to limit reverse causality, we excluded the values of energy intake in the last 2 years prior to death, which resulted in excluding 77 participants who died in the first 2 years. Moreover, due to the potential effect of cancer treatment on dietary intake and body weight, we excluded people who died from non-cancer causes but developed a cancer during the follow-up (*N* = 12). Therefore, the baseline sample was of 7119 participants, of which, 239 died up to 1st December 2010. The change over time analysis included only 6180 with valid EI data at least at one follow-up time point. The study flowchart is available in Figure 1. The STROBE checklist is available in Supplemental Appendix A.

### 2.2. Outcome Variables

The main outcome was all-cause mortality, for which cases up to 1st December 2010 were determined by the Clinical Event Committee through follow-up study visits, periodic review of medical records, repeated contact with the participants, and linkage to the national death registry [13,14]. The cause of death was registered and classified as of cardiovascular, cancer, or another cause. We used cardiovascular and cancer mortality as secondary outcomes. The time-to-event was calculated as the time from the baseline visit to date of death, or 1 December 2010 for non-cases (censoring).

### 2.3. Exposure Variables

At baseline and at each yearly visit, participants completed a validated 137-item food frequency questionnaire [16,17], from which we estimated energy intake in kcal/day. The main exposure was the proportion of energy requirement covered by energy intake. At each visit, the theoretical energy expenditure (TEE) was calculated according to the following equations [18]: for men, TEE = 864 − 9.72 × age (years) + PA × ((14.2 × weight (kg) + 503 × height (meters)) and for women: TEE = 387 − 7.31 × age (years) + PA × ((10.9 × weight (kg) + 660.7 × height (meters)), where PA is a physical activity coefficient. The values of PA for men was 1, 1.12, 1.27, 1.54 and for women 1, 1.14, 1.27, 1.47, which were attributed to those in the first, second, third and fourth quartile of leisure time physical activity, respectively [19]. Then, at each time point, the difference between the energy consumed and theoretical requirement was calculated as 100 × (EI–TEE)/TEE, so that, for example, if a participant reports a daily EI of 2000 kcal and their TEE is of 1800, they consume an excess (2000 − 1800)/1800 = 11% of energy compared to their theoretical requirement. Conversely, for example, if their EI is 1500 kcal and their TEE is 1800, they are in energy deficit of −(1500 − 1800)/1800 = 17%. Many participants did not attend all the yearly visits; therefore, the number of times the participants were interviewed varies from one participant to another. The cumulative average was calculated as the sum of these values at each visit with available data until the last visit before censoring or 2 years before death, divided by the number of time points with non-missing data. For 939 individuals, only the baseline value was available. For the rest of participants (*N* = 6180), the exposure was the cumulative average over at least 2 time points.

The secondary exposure was a change in energy intake from baseline, expressed in percentage of the baseline value. At each follow-up visit (t) with available energy data, a change in energy intake from baseline (t0) was calculated as follows: 100 × (energy intake t − energy intake t0)/energy intake t0. The average of the changes was calculated as the sum of changes until the last visit before censoring or 2 years before death divided by the number of time points. This was available for *N* = 6180 participants who had at least one follow-up value to calculate the change from baseline.

### 2.4. Covariates

Trained personnel collected baseline data on age; sex; educational level; prevalence of diabetes, hypercholesterolemia, hypertriglyceridemia, and hypertension; systolic blood pressure; body mass index; and smoking habit [13,14]. To account for diet quality independently of energy intake, we used a Mediterranean diet adherence score, a validated 14-item short screener on the essential characteristics of a Mediterranean diet (favorable items: fruits, vegetables, legumes, extra-virgin olive oil, mixed nuts, fish, wine in moderation; detrimental items: animal fats, red and processed meats, processed foods, and sugary drinks) [20]. We calculated the cumulative average throughout the follow-up to account for improvement in Mediterranean diet adherence over time, particularly as a result of the intervention.

Leisure time physical activity was estimated by the self-administered Minnesota Leisure-Time Physical Activity Questionnaire [21,22]. The questionnaire reported the number of days and min/day they performed 67 different activities in the previous year. Leisure-time physical activity was quantified in metabolic equivalents of task-min/day by multiplying the metabolic equivalents of a task linked to an activity with its mean duration reported in min/day.

### 2.5. Statistical Analysis

The primary outcome was death of all-cause, and secondary outcomes were cardiovascular mortality and cancer mortality. Cox proportional hazard models were fitted with the exposure mid-term energy requirement covered by energy intake. This was modelled as a continuous predictor, and as binary variables (yes/no) according to the following cut-off points: an energy surplus of at least 20%, 25%, 30% or 35%, or energy deficit of at least 20%, 25%, 30% or 35% below the requirement. For the categorical analyses, the comparison group was the “no” category; for example, for the variable “energy surplus > 20%”, we compared people with an energy surplus of at least 20% above requirements to people with an energy intake up to a surplus of 20%. The models were stratified by sex, study center, and educational level, and adjusted for age, intervention group, hypertension, hypertriglyceridemia, hypercholesterolemia, diabetes, Mediterranean diet score, alcohol intake, smoking status. As weight, height and physical activity are comprised in the calculation of TEE, the models did not include body mass index and physical activity to avoid over-adjustment.

A similar modelling approach was used with the cumulative average of change in energy intake, as a percentage of baseline energy intake: continuous and in the following categories: a reduction in energy intake compared to baseline of 20% or more, 25% or more, 30% or more or 35% or more, or an increase in energy intake compared to baseline of at least 20%, 25%, >0% or 35%. The models included body mass index and physical activity at baseline as covariates.

We performed three sets of sensitivity analyses. (1) We restricted the sample to participants with at least two time points (*N* = 6180) for the analysis of energy balance. (2) We limited the reference group for each analysis to participants with an energy balance within the “normal range”, to avoid having a reference group of participants with extreme energy intake which may blur the interpretation. For the analyses of “energy surplus”, we excluded participants with very low energy intake from the reference group (consuming less than 25% below requirement). Similarly, for “reduction in energy intake”, the reference groups excluded participants with a reduction of 25% or more. To mirror this, for analyses of “energy deficit” we excluded participants consuming at least 25% over the requirement from the reference category, and for analyses of “reduction in energy intake”, we excluded participants with a change >+25%. (3) To investigate the potential mediating role of cardiovascular risk factors (diabetes, hypertension, dyslipidemia) and of weight change in the observed associations between energy balance and mortality, we fitted nested models, gradually adding these covariates.

We accepted any two-sided *p*-value < 0.05 as significant, and ran the analyses in SAS 9.4 (Cary, NC, USA).

## 3. Results

Among the 7119 participants, after a median follow-up of 4.80 years, there were 239 death cases occurring after 2 years, of which 103 were from cancer and 57 from cardiovascular disease. The participants’ characteristics at baseline are presented in Table 1. Compared to people who did not die, participants who died during follow-up were older, more likely to be men, current smokers, had poorer adherence to a Mediterranean diet, and were more likely to have an energy intake above their energy needs by more than 30% (*p* < 0.001). Compared to people who had at least one follow-up datum on energy intake, participants with only baseline data available had a lower Mediterranean diet score, lower physical activity levels, lower total energy and alcohol intake, a slightly higher BMI and were more likely to have diabetes (Appendix A).

As presented in Figure 2, a cumulative average of energy intake exceeding energy needs was associated with an increase in mortality risk: continuous hazard ratio (HR) for a 10% energy surplus = 1.10; 95% confidence interval 1.02, 1.18. This was also the case for cardiovascular disease mortality (continuous HR_10% energy excess_ = 1.26; 95% CI 1.11, 1.43), but less apparent for cancer mortality. Consistently consuming energy below one’s needs was not associated with a lower risk. Rather, an extreme energy underconsumption was associated with greater all-cause mortality, likely driven by cancer mortality, for which double the risk of death was observed: HR_energy 35% below energy requirement_ = 2.04; 95% CI: 1.12, 3.74. In a restricted sample of participants with available data on energy intake at least at one follow-up visit (*N* = 6180, Appendix A), results were essentially similar for all-cause mortality (continuous HR_10% energy excess_ = 1.12; 95% CI 1.02, 1.23) and CVD mortality (continuous HR_10% energy excess_ = 1.25; 95% CI 1.01, 1.55) with wider confidence intervals. In that sample, there was no evidence of an excess cancer mortality at extreme energy deficit (HR_energy 35% below energy requirement_ = 1.55; 95% CI: 0.66, 3.65). When restricting the reference category to people with an energy intake within the “normal range”, results were essentially the same (Appendix A).

Among the 6180 participants with data on energy change compared to baseline, there were 139 cases of death, of which 60 were caused by cancer and 32 by cardiovascular disease. A continuous association between a change in energy intake and all-cause mortality (continuous HR_10% increase_ = 1.09; 95% CI 1.02, 1.17) was observed, suggesting that an increase over time of energy intake was associated with slightly greater hazards of death of any cause (Figure 3). There was no evidence that reducing energy intake over time by 20% or more would reduce all-cause mortality risk. The results were similar when restricting the reference category to participants with a relatively steady energy intake (energy change compared to baseline within −25% and +25%), as seen in Appendix A.

The observed associations with both energy balance and change in energy intake did not appear to be mediated neither by cardiovascular risk factors, nor change in body weight over time. Estimates were similar in models without adjustment for BMI, adjusted for baseline BMI only and adjusted for both baseline and change in BMI (Appendix A).

## 4. Discussion

In this large cohort of Spanish older adults at high cardiovascular risk, we observed that a sustained energy surplus over 5 years and an increase in energy intake compared to baseline are associated with a higher risk of mortality. Energy deficit or a decrease in energy intake over time did not appear to be associated with a lower mortality risk.

The results we report on the association between energy surplus and higher mortality risk are consistent with the extensive body of literature showing that excess weight and adiposity are associated with increased risk of mortality [1,2,23]. However, it is important to note that the association with energy surplus that we observe was independent of body size, and that change in body mass index did not appear to be a mediator of the association. This indicates that a sustained surplus of energy compared to requirements can have metabolic consequences beyond its effects on body size, including low-grade inflammation and oxidative stress [24]. Only a few studies investigated the association of energy intake with mortality, independent of body mass index [12,25]. In the Malmö study [12], a non-significant “U-shaped” association was reported, with a lower risk of all-cause mortality observed in the middle quartiles of energy intake after 6.6 years of follow-up and after the exclusion of individuals with <1 year of follow-up. The shape of the association is consistent with the results we report in the present study, although our results present a much clearer association between high energy intake (in excess compared to the individual’s requirement, or an increase over time) with elevated mortality risk. The Malmö study did not adjust for overall diet quality and the authors discuss that reverse causality may explain their results, as people with low energy intake may present diminished energy demand and decreased appetite, existing chronic conditions, or have a low-quality diet. In a study in middle age and elderly Japanese adults followed-up for 29 years [25], higher energy intake was associated with greater mortality risk, although only in men, after adjusting for a wide set of confounders, including diet quality. Our results align with those of the Japanese study, although they found an association of the similar magnitude for cancer and cardiovascular disease mortality, whereas in our study, the association between energy surplus and mortality was driven by deaths of cardiovascular cause.

The observed excess risk in cancer mortality at extreme energy deficit (at least 35% below energy requirements) was likely due to reverse causality. When restricting the sample to people with at least one follow-up measurement, which is a measurement of greater precision, also indicating no early dropout, this association did not hold.

Despite being supported by animal data, including non-human primates followed-up for over 20 years [4,26], the theory that energy deficit can reduce premature mortality was not supported by our results. The CALERIE trial was a 2-year 25% calorie restriction intervention study in non-obese individuals aged 21–51 years [27]. It was found that the calorie restriction group had larger decreases in cardiometabolic risk factors and some markers of longevity than the control group, adjusted for weight change. Mortality was not an outcome of this trial conducted in healthy young individuals; therefore, it cannot be compared to our study. A major difference is the interventional nature of this study, while in the PREDIMED trial, the intervention diet was not calorie-restricted, and here, we used an observational design adjusting for intervention group. Moreover, participants in the present study were 55–80 years old with metabolic syndrome. Therefore, one explanation might be the time window at which energy deficit is exerted, older age being already “too late” to reduce mortality risk.

This study has several strengths. The main originality is the repeated yearly measurements of energy intake, physical activity, body weight and height and diet quality that allowed a fine estimation of changes over time and of sustained energy excess or deficit and covariates. It was also the first study to use as an exposure the adequation of energy intake to energy requirements depending on age, body mass and physical activity levels, as all these data were available at every yearly visit, and anthropometrics were measured by a nurse at each of these visits. Mortality is a robust outcome that was ascertained via various methods (contact, revision of medical records, linkage with death records). The results were robust to various levels of adjustment, of choice of reference category and of restriction of the sample with more available data. There are also some important limitations to point out. Firstly, the information on energy intake was derived from a food frequency questionnaire, which is an instrument that requires literacy and cognitive abilities. As it is true for all self-reported dietary assessment methods, FFQs are subject to measurement errors [28], due to different reasons, including memory, social desirability, misestimation of serving sizes and frequencies, and overall difficulties to synthetize, as well as incorrect nutrient estimates from the grouped foods included in the questionnaires. Self-reported energy intake is inaccurate by essence [29], and it has been shown that this questionnaire tends to overestimate energy intake compared to 3 days of diet recording [17]. However, the questionnaires were administered in person by trained dietitians, and they have repeatedly shown a sufficient degree of validity in epidemiological studies and in specific studies of validation and reproducibility [16,17]. Similarly, physical activity was estimated by questionnaire and not by objective methods such as accelerometers, but this questionnaire has been widely used and validated in the Spanish population [21,22]. Secondly, the relatively short-term follow-up may be subject to reverse causation, such as the fact that individuals who are chronically ill may decrease their overall food intake for several years before death. We tried to avoid this by excluding data on energy intake during the 2 years prior to death, as a balance between epidemiological rigor and keeping statistical power, but reverse causation may still have occurred. Thirdly, since the original PREDIMED trial did not restrict energy intake [13], all the PREDIMED participants should be considered ad libitum in terms of energy intake, and the term of energy deficit in this observational study is obviously not as drastic as the conditions of extreme energy restriction such as a famine. However, we observe that a substantial proportion of the participants (11%) have an energy intake of at least 30% below their energy requirement, making the analysis of moderate energy deficit possible. Finally, the study sample (middle-aged/elderly individuals at high cardiovascular risk) limits the generalizability of our results to other populations.

To conclude, based on a food frequency questionnaire and estimation of energy requirements by equations, consuming energy in excess compared to an individual’s energy requirements, or a sustained increase in energy intake over time were associated with a higher risk of mortality in this cohort of Spanish older adults over 5 years of follow-up. Moderate energy deficit, or a reduction in energy intake did not appear to be associated with lower risk of mortality. The follow-up of intervention trials designed to induce long-term energy reduction in older adults, such as the PREDIMED-Plus trial, will help corroborate or contradict these results.

## Figures and Tables

**Figure 1 nutrients-13-01545-f001:**
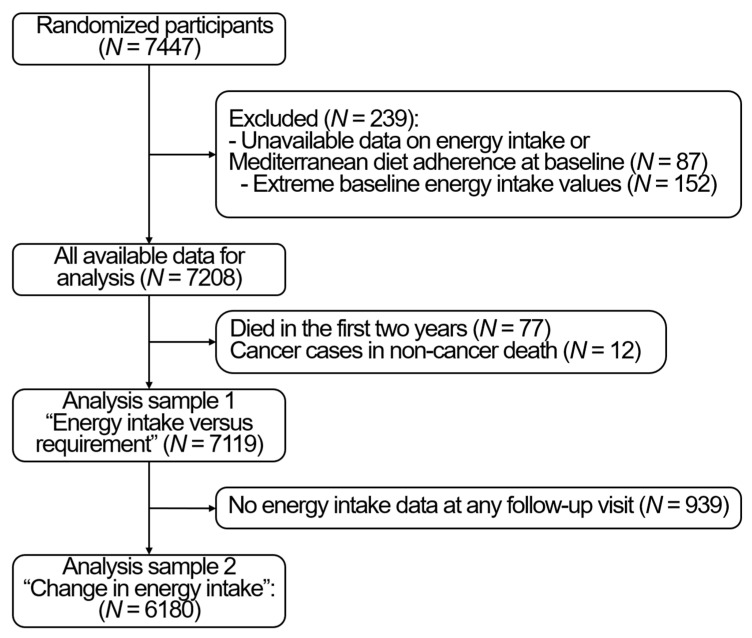
Study flow chart.

**Figure 2 nutrients-13-01545-f002:**
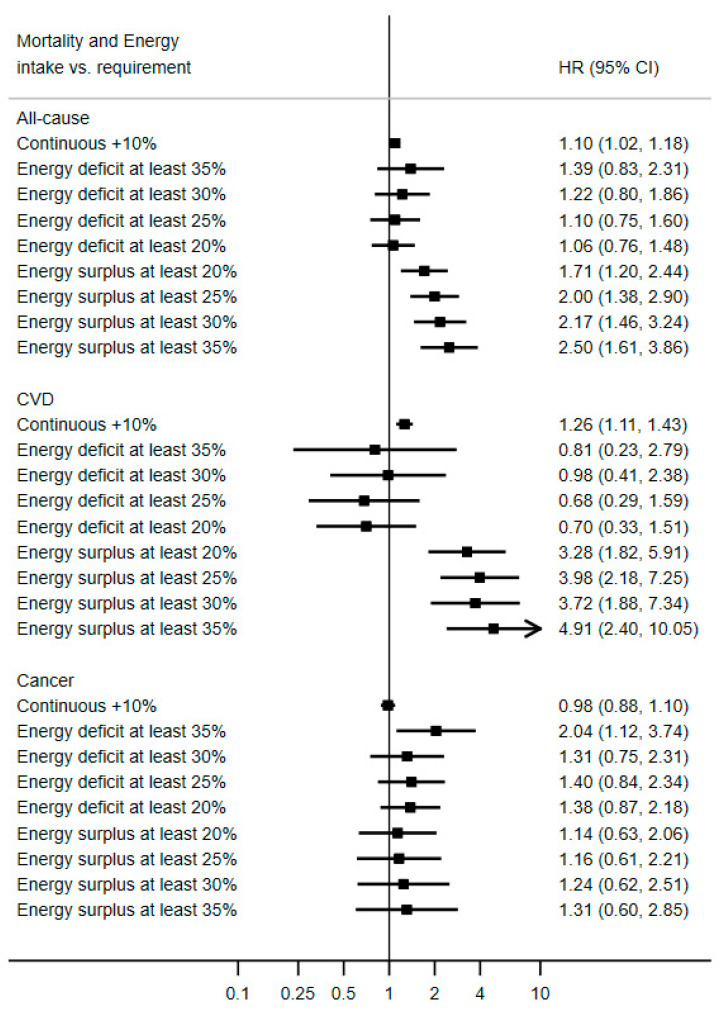
Association between long-term ratio of energy intake to theoretical energy requirement and mortality risk, PREDIMED study, *N* = 7119. Values are multivariable hazard ratios (HRs) and 95% confidence intervals, stratified by sex, study center and education level, and adjusted for baseline age, intervention group, hypertension, hypertriglyceridemia, hypercholesterolemia, diabetes, alcohol intake, smoking status and cumulative average of Mediterranean diet score. Energy deficit is defined as energy intake below energy requirement. Energy surplus is defined as energy intake above energy requirement.

**Figure 3 nutrients-13-01545-f003:**
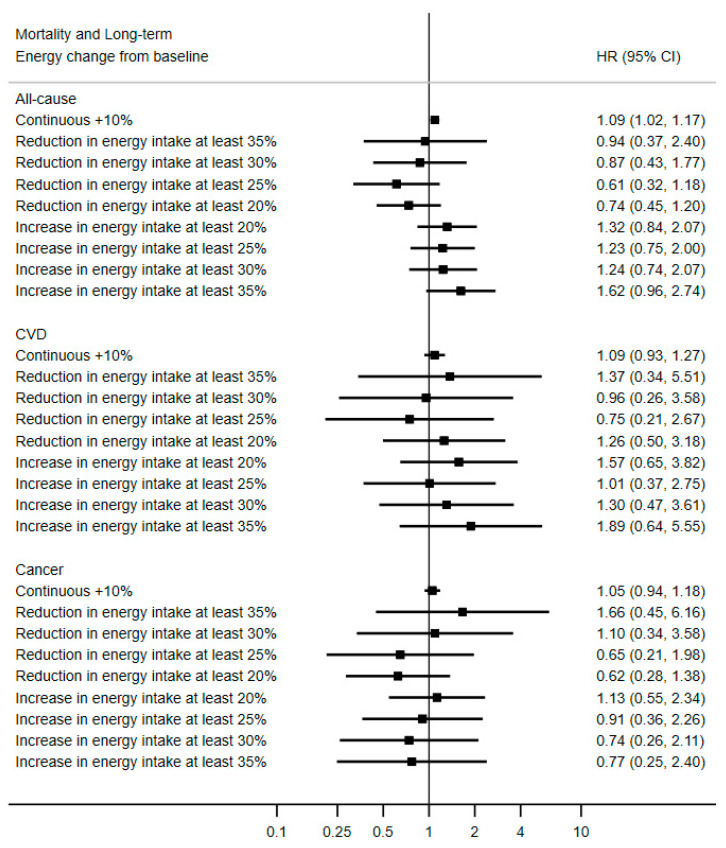
Association between long-term change in energy intake compared to baseline and mortality, PREDIMED Study, *N* = 6180. Values are multivariable hazard ratios (HRs) and 95% confidence intervals, stratified by sex, study center and education level, and adjusted for baseline age, intervention group, hypertension, hypertriglyceridemia, hypercholesterolemia, diabetes, alcohol intake, smoking status and cumulative average of Mediterranean diet score, baseline physical activity and body mass index.

**Table 1 nutrients-13-01545-t001:** Baseline characteristics of the participants included in the analysis sample 1 (*N* = 7119), by mortality status, PREDIMED study.

	Non-Cases	Mortality Cases	*p*-Value ^a^
*N*	6880	239	
Women, *n* (%)	4015 (58.4)	97 (40.6)	<0.0001
Age, y	66.9 (6.1)	70.6 (6.5)	<0.0001
Energy intake baseline kcal, mean (SD)	2234 (541)	2270 (577)	0.31
Energy intake vs. requirement %, median (IQR)	−5.3 (−19.6; 11.1)	−3.8 (−21.7; 14.9)	0.14
Energy intake at least 30% below requirement, *n* (%)	760 (11.1)	30 (12.5)	0.47
Energy intake at least 30% above requirement, *n* (%)	583 (8.5)	36 (15.1)	<0.0001
Change in energy intake compared to baseline (%), median (IQR)	−2.1 (−15.8; 14.0)	+2.8 (−14.7; 12.2)	0.32
Reduction in energy intake at least 30% compared to baseline, *n* (%)	433 (7.2)	9 (6.5)	0.75
Increase in energy intake at least 30% compared to baseline, *n* (%)	662 (11.0)	16 (11.5)	0.84
Mediterranean diet adherence score (0–14), mean (SD)	9.59 (1.58)	9.29 (1.58)	0.004
Physical activity METs min/day, mean (SD)	231.5 (238.8)	215.7/216.2)	0.27
Alcohol intake g/day, mean (SD)	8.2 (13.8)	10.6 (18.7)	0.05
BMI kg/m^2^, mean (SD)	30.0 (3.8)	29.7 (3.9)	0.25
Hypertension, *n* (%)	5692 (82.7)	197 (82.4)	0.90
Hypercholesterolemia, *n* (%)	5010 (72.8)	133 (55.7)	<0.0001
Hypertriglyceridemia, *n* (%)	1957 (28.4)	78 (32.6)	0.16
Current smokers, *n* (%)	1653 (24.0)	79 (33.0)	<0.0001
Diabetes, *n* (%)	3317 (48.2)	152 (63.6)	<0.0001

^a^ *p*-value of the *t*-test (for continuous variables) or Chi square (for categorical variables) test of the difference between non-cases and cases of mortality; abbreviations: SD, standard deviation; IQR, interquartile range.

## Data Availability

The datasets analyzed in the current study are not publicly available due to data regulations and for ethical reasons, considering that this information might compromise research participants’ consent because our participants only gave their consent for the use of their data by the original team of investigators. However, collaboration for data analyses can be requested by sending a letter to the PREDIMED steering Committee (predimed-steering-committe@googlegroups.com). The request will then be passed to all the members of the PREDIMED Steering Committee for deliberation.

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
