# Peer review of "Energy Balance and Risk of Mortality in Spanish Older Adults"

_nutrients, 2021, doi:10.3390/nu13051545_

Round 1

Reviewer 1 Report

Manuscript #: Nutrients (ISSN 2072-6643)

Title: Energy balance and risk of mortality Spanish older adults.

General: This study examined the effect of sustained caloric restriction on 5-year mortality; and the effect of energy intake (EI) on all cause, cardiovascular and cancer mortality among participants from the PREvención con DIeta MEDiterránea Study.

Abstract: Comment:

-Please have the same study objective in the abstract and in the introduction. The study objective in the introduction section is clearer than the one in the abstract.

Introduction: The literature used is pertinent to the study. The study objective was clearly stated.

Methods: Comments:  

-Please provide information on how many times the participants were interviewed.

-How different or similar were the 939 participants that did not have any data at follow-up when compared with those included in the analysis. How they have impacted the actual findings if they were included?

Results: Comments:

-I do not see findings presented by stratification (sex, study center, and education level) as it is stated at the bottom of each figure. Please clarify.

Discussion: Previous pertinent literature was compared with author’s findings. Study strengths and limitations were identified.

Comment:

-What are the study implications?

References: All appropriate.

Author Response

Manuscript #: Nutrients (ISSN 2072-6643)

Title: Energy balance and risk of mortality Spanish older adults.

General: This study examined the effect of sustained caloric restriction on 5-year mortality; and the effect of energy intake (EI) on all cause, cardiovascular and cancer mortality among participants from the PREvención con DIeta MEDiterránea Study.

Abstract: Comment:

-Please have the same study objective in the abstract and in the introduction. The study objective in the introduction section is clearer than the one in the abstract.

Our response: Thank you, we have now changed the objectives in the Abstract (line 35) as follows: “We aimed to assess the association with all-cause, cardiovascular and cancer mortality of 1) sustained caloric restriction, calculated according to each individual’s energy intake (EI) and theoretical energy expenditure (TEE), and 2) mid-term change in total EI in a prospective study.”

Introduction: The literature used is pertinent to the study. The study objective was clearly stated.

Our response: Thank you.

 Methods: Comments:  

-Please provide information on how many times the participants were interviewed.

Our response: There was the initial interview at baseline, then at every follow-up visit which happened every year. We have added the word “yearly” on line 106 to clarify this, and we also added “At baseline and at each yearly visit” on line 126. However, many participants did not attend all the yearly visits, sometimes they would miss one year but then be interviewed the following year, etc. Therefore, the number of times the participants were interviewed varies from one person to another. We have clarified this on line 138: “Many participants did not attend all the yearly visits, therefore the number of times the participants were interviewed varies from one participant to another. The cumulative average was calculated as the sum of these values at each visit with available data until the last visit before censoring or 2 years before death, divided by the number of time points with non-missing data. For 939 individuals, only the baseline value was available. For the rest of participants (N=6180), the exposure was the cumulative average over at least 2 time points.”

-How different or similar were the 939 participants that did not have any data at follow-up when compared with those included in the analysis. How they have impacted the actual findings if they were included?

Our response: Thank you for this important remark. We have now added in Supplemental table 1 the comparison of the 6180 participants with follow-up data to the 939 without follow-up data. We comment on these differences in the Results, line 210: “Compared to people who had at least one follow-up data on energy intake, participants with only baseline data available had a lower Mediterranean diet score, physical activity levels lower total energy and alcohol intake, a slightly higher BMI and were more likely to have diabetes (Supplemental Table 1).”

Regarding the potential impact on findings, for the analyses responding to objective 1), we have conducted a sensitivity analysis by excluding the 939 participants that did not have any data at follow-up , and the results were essentially the same, as described on line 222-228: “In a restricted sample of participants with available data on energy intake at least at one follow-up visit (N=6180, Supplemental Figure 1), results were essentially similar for all-cause mortality (continuous HR10% energy excess =1.12; 95% CI 1.02, 1.23) and CVD mortality (continuous HR10% energy excess =1.25; 95% CI 1.01, 1.55) with wider confidence intervals. In that sample, there was no evidence of an excess cancer mortality at extreme calorie restriction (HR energy <-35% energy requirement = 1.55; 95% CI: 0.66, 3.65).”

Results: Comments:

-I do not see findings presented by stratification (sex, study center, and education level) as it is stated at the bottom of each figure. Please clarify.

Our response: Thank you for this remark. The term “stratified” here does not refer to a subgroup analysis (by sex, by study center, by educational level). Rather, it is the modelling strategy of the Cox proportional hazards regression, that allows not only to adjust for covariates, but also to stratify, which estimates a different baseline hazard for the different strata (e.g. for men and women).

Discussion: Previous pertinent literature was compared with author’s findings. Study strengths and limitations were identified.

Comment:

-What are the study implications?

Our response: As an observational study, there is no direct clinical implication of our results, that need to be replicated in other studies, in particular long-term intervention trials specifically designed to induce sustained calorie restriction. We have added the following sentence on line 347: “Follow-up of intervention trials designed to induce long-term calorie reduction in older adults such as the PREDIMED-Plus trial, will help corroborate or contradict these results.”

References: All appropriate.

Reviewer 2 Report

Comments for Authors

I read with interest your article “Energy balance and risk of mortality in Spanish older adults”. Your study report aimed to assess the association of sustained energy restriction or sustained energy excess over a median period of 4.8 years in participants at high risk of developing cardiovascular disease.

The strengths of this article are it reports on a large prospective study that was well-designed, well-resourced and the results were analysed using appropriate methodology. There were measures taken at each stage of the study and in the analysis of the data to ensure quality data was collected, appropriate participant results were included and appropriate interpretations of the results were reported in the article. I was particularly impressed to see dietitians provided the appropriate diet education to the study participants in each arm of the study and the dietitians also administered the food frequency questionnaires in person to improve the quality of the data collected.

I found the use of the less than symbol (<) confusing in Figure 2 and Figure 3 on pages 6 and 7 of the paper.

It seems ”caloric restriction <35%” in Figure 2 is meant to represent a caloric restriction of at least 35% below current energy requirements. Unfortunately to me on initial readings I read it as up to 35% below current requirements. For clarity and easy understanding of the results in Figure 2 and Figure 3, I hope the authors will consider avoiding use of the symbols (< and >) in the Figures 2 & 3 and replacing the symbols with clear wording.

Author Response

I read with interest your article “Energy balance and risk of mortality in Spanish older adults”. Your study report aimed to assess the association of sustained energy restriction or sustained energy excess over a median period of 4.8 years in participants at high risk of developing cardiovascular disease.

The strengths of this article are it reports on a large prospective study that was well-designed, well-resourced and the results were analysed using appropriate methodology. There were measures taken at each stage of the study and in the analysis of the data to ensure quality data was collected, appropriate participant results were included and appropriate interpretations of the results were reported in the article. I was particularly impressed to see dietitians provided the appropriate diet education to the study participants in each arm of the study and the dietitians also administered the food frequency questionnaires in person to improve the quality of the data collected.

Our response: We thank the reviewer for their very enthusiast and positive appraisal of our manuscript.

I found the use of the less than symbol (<) confusing in Figure 2 and Figure 3 on pages 6 and 7 of the paper.

It seems ”caloric restriction <35%” in Figure 2 is meant to represent a caloric restriction of at least 35% below current energy requirements. Unfortunately to me on initial readings I read it as up to 35% below current requirements. For clarity and easy understanding of the results in Figure 2 and Figure 3, I hope the authors will consider avoiding use of the symbols (< and >) in the Figures 2 & 3 and replacing the symbols with clear wording.

Our response: Thank you, we have replaced the symbols by the wording “calorie restriction at least 35%”, “calorie excess over 35%” on Figure 2 and by “energy change at least -35%” and "energy change at least +35%" on Figure 3.